

# Significant Influence of UV-vis Irradiation on Cloud Activation Efficiencies of Ammonium Sulfate Aerosols under Simulated Chamber Conditions

Anil Kumar Mandariya[1], Junteng Wu[2, +], Anne Monod[2], Paola Formenti[3], Bénédicte Picquet-Varrault[1], Mathieu Cazaunau[1], Stephan Mertes[4], Laurent Poulain[4], Antonin Berge[3], Edouard Pangui[1], Andreas Tilgner[4], Thomas Schaefer[4], Liang Wen[4, ++], Hartmut Herrmann[4], and Jean-François Doussin[1]

[1] Univ Paris Est Creteil and Université Paris Cité, CNRS, LISA, F–94010 Créteil, France

[2] Aix Marseille Université, CNRS, LCE, Marseille, France

[3] Université Paris Cité and Univ Paris Est Creteil, CNRS, LISA, F–94010 Créteil, France

[4] Leibniz Institute for Tropospheric Research, Leipzig (TROPOS), 04318, Germany

+ : now: at Laboratoire de Météorologie Physique, UMR 6016, CNRS, Université Clermont Auvergne, 63178 Aubière, France

++ : now: at Chinese Research Academy of Environmental Sciences (CRAES), Beijing 100012, China

*Correspondence to*: Jean-François Doussin (Jean-Francois.Doussin@lisa.ipsl.fr)

**Abstract:** In this work, an optimized protocol to generate an expansion-type liquid clouds with and without UV-visible light irradiation conditions for simulation chamber studies is presented. Sensitivity of the process to key parameters such as initial relative humidities, temperature inhomogeneities, droplets lifetime or seed particle number is illustrated. The obtained clouds have shown that not all seeds particles were always activated and so an iterative numerical method has been re-designed to separate cloud droplets from non-activated seed particles during data analysis allowing the characterization of the cloud droplet formation properties without CCN counter data.

Two types of experiments, clouds without irradiation (N-IC) and under UV-visible light irradiation conditions (IC), have been conducted in the CESAM multiphase atmospheric chamber. Measured cloud droplet lifetimes were in good agreement with atmospheric droplet lifetimes. The achieved supersaturation in the cloud was mostly sensitive to the initial relative humidity in the chamber. The comparison between the cloud formation pattern of N-IC and LC was also investigated. Under illumination conditions, the generated clouds clearly showed a gradual activation of seed particles into droplets and thus of the microphysical properties like LWC and droplet concentration, while under dark conditions, clouds faced a flash activation of seed particles. Because this phenomenon may also impact the air/water partitioning of semi-volatile compounds, and it should be considered for further studies, especially in further multiphase photochemical studies implying water-soluble volatile organic compounds in the CESAM chamber.





## 1 Introduction

Atmospheric aerosol particles, acting as cloud condensation nuclei (CCN), affects the formation, as well as the microphysical and radiative properties od clouds (Martinsson et al., 1999; Twomey, 1959; McFiggans et al., 2006), still one of the major uncertainties in the attribution of climate forcing (Intergovernmental Panel on Climate Change, 2023). The process of CCN (also named seed particles in chamber experiments) activation into cloud droplet is often addressed as "cloud activation". The cloud droplet size is controlled mainly by the local meteorological parameters and physicochemical properties of CCN. CCN activation into droplet requires a necessary amount of water (critical supersaturation) and depends in a complex way on cooling rate, aerosol particle size, and chemical composition (Twomey, 1959).

Pruppacher (1986) reported that more than 90% of the atmospheric clouds on Earth re-evaporate without precipitation, implying that a CCN particle is processed through several non-precipitating cloud life cycles before being removed through precipitation. At the same time, a cloud droplet can absorb water-soluble gases, including volatile organic compounds (VOCs), and oxidants. This reactive mixture can form less volatile compounds that may remain as residues in a CCN after droplet evaporation. As a consequence turn, the cycles of formation-evaporation of non-precipitating clouds have the potential of increasing the aerosol mass as well as of altering the physicochemical properties of CCN (Brégonzio-Rozier et al., 2015; Ervens et al., 2011; Giorio et al., 2017; De Haan et al., 2017; Hoyle et al., 2016a; Mertes et al., 2005a, 2005b).

In general, uncertainties still exist as these processes on or in cloud droplets are poorly understood under dark and light conditions. Similarly, understanding of cloud microphysics in simulation chamber under both dark and light conditions needs improved and controlled, which would be crucial to investigate the aqueous secondary organic aerosol (aqSOA) formation (Lim et al., 2013) or the aging through cloud processing of already existing aerosol. However, it is challenging to investigate the actual cloud droplets in the atmosphere because clouds are highly complex and usually occur at inconvenient locations with sporadic and nonstationary occurrences (Stratmann et al., 2004). Therefore, laboratory investigations using cloud and multiphase atmospheric chambers in conditions relevant to the atmosphere are henceforth key to better understand and quantify cloud formation properties, as well as the formation and aging of the organic aerosol (OA) during cloud-formation-evaporation cycles (Kreidenweis et al., 2019; Stratmann et al., 2009). However, these experiments need to be reproducible and understood to provide with meaningful results.

Over the last few decades, various cloud and multiphase simulation chambers, namely DRI chamber (Stehle et al., 1981), CALSPAN (Hoppel et al., 1994), AIDA (Möhler et al., 2001), AIDAd (Alpert et al., 2023), LACIS (Stratmann et al., 2004), CLOUD (Duplissy et al., 2010), CESAM (Wang et al., 2011), MRI (Tajiri et al., 2013), MICC (Frey et al., 2018) and the Pi chamber (Chang et al., 2016), were used to investigate the cloud microphysical process and cloud life cycles, chemical transformations inside and at the droplet interface. Among these, expansion-type cloud chambers were used to generate clouds by performing a quasi-adiabatic expansion through a decrement of chamber pressure with or without controlling the wall temperature. This method generates a few minutes of long liquid clouds (nearly equal to atmospheric cloud droplet lifetime) and mixed-phase clouds (liquid and ice) clouds. The cloud lifetime is defined by the time during which suspended droplets are observed in the chamber. However, the cooling rate varies from high to low depending on the chamber type. Tajiri et al. (2013) induced a dark liquid cloud on ammonium sulfate seed particles (80 nm mode diameter) by active pumping from



1000 to 850 mbar with an adiabatic ascent rate of 3 m s$^{-1}$. They reported that seed particles started activating into
cloud droplets after 4-min pumping, and nearly 70% activated at 1% supersaturation. Frey et al. (2018) generated
a dark expansion cloud on ammonium seed particles containing organic compounds and observed an unexplained
flash activation of seed particles into droplets just after a minute of pumping, subsequently decreasing the number
concentration of droplets. The seed particle activation ratio (for liquid cloud droplets) is defined as the fractional
activation of seed aerosol particles into cloud droplets. This seed particle activation ratio can depend on turbulence,
as turbulence induces a fluctuation in the supersaturation ratio (Shawon et al., 2021). Abade et al. (2018) suggested
that some "fortunate" CCN particles might get activated into droplets because of this fluctuation. These
supersaturation fluctuations lead to an increment in the seed particle activation ratio, and also broadens the cloud
droplet size distribution (Prabhakaran et al., 2020). Further, the cloud droplet formation is predominantly
controlled by the number concentration of CCN particles (Hoyle et al., 2016b). The droplet activation ratio
decreases monotonically as the concentration of CCN increases (Shawon et al., 2021).
All these studies mainly focused on chemical and microphysical transformations of aerosols and microphysical
properties of ice and mixed-phase clouds, turbulent clouds, and cloud processing of secondary organic aerosols
(SOA). None of the investigations listed above investigated the microphysical properties of a liquid cloud
generated by a quasi-adiabatic expansion under dark and simulated light conditions relevant to the atmosphere.
Although extremely challenging, the control of cloud formation under dark and light conditions is necessary for
further multiphase photochemical studies under realistic conditions.  In this paper, we present a study aiming to
optimize a controlled protocol for generating quasi-adiabatic expansion clouds of realistic liquid droplets under
atmospheric relevant simulated dark and light conditions. Detailed microphysical characterizations of these clouds
using monodispersed ammonium sulfate seed particles were performed in the CESAM chamber. The experiments
were carried out under the PARAMOUNT project at the CESAM chamber as a basis for further inquiries on cloud
assisted SOA formation/evolution that will be described in future papers.
**2 Experimental Section**
The CESAM atmospheric chamber, described in detail by Wang et al. (2011) and Brégonzio-Rozier et al. (2015);
is a vacuum-compatible 4.2 m$^3$ cylindrical stainless-steel reactor equipped with three Xenon arc lamps (3 × 6500
W) and Pyrex filters of 6.5 mm thickness. These lamps and filters produce an irradiation spectrum very similar to
the ground-level solar spectrum, both in terms of intensity and spectral distribution. CESAM is a double-walled
reactor, temperature-controlled thanks to a coolant circulating inside the walls.
Cloud generation under nearly atmospheric conditions is extremely challenging, and therefore, the experimental
protocols have been optimized to get close to realistic liquid droplet cumulus/liquid clouds for approximately 10
min, considered as 10 min pumping. The chamber was filled with a mixture of N$_2$/O$_2$ at 80/20% (generated using
N$_2$ from liquid nitrogen evaporation, purity >99.995%, H$_2$O<5ppm, Messer, and O$_2$, quality N5.0, purity> 99.995
%, H$_2$O < 5 ppm, Air Liquide). The same N$_2$/O$_2$ mixture was also used to compensate from sampling by various
instruments and maintain a constant pressure in the reactor. Three large beakers (height: 40 cm; diameter: 25 cm)
were placed between the top of the chamber and the lamps, as shown in Fig. 1, to absorb infrared radiation from
the light and prevent from overheating while irradiating the sampling volume. These beakers were filled nearly
half (up to 20-25 cm) with water which was renewed before each experiment. CESAM is connected to a vacuum





system consisting of two pumping circuits. A first pumping line served to evacuate the air at 100 L min⁻¹ during
each cloud run. It consisted in a dry, oil-free screw vacuum pump (Bush® CobraTM N0100–0300B) supported by
a root pump (Leybold® RUVACTM WAU 501) mounted on its forehead. The second pumping line served to clean
the chamber in between experiments by creating a vacuum in the $10^{-4}$ mbar range, and consisted in a
turbomolecular pump (Leybold® Turbovac 361®). For the cloud runs, the evacuation rate was precisely controlled,
by means of a high flow mass controller (ALICAT SCIENTIFIC).
A small, 5 l stainless steel vessel is installed below the chamber to generate pressurized water vapor. In addition,
a glass balloon and round bottom borosilicate flask were also connected to the chamber for vacuum water vapor
injection and water vapor compensation, as shown in Fig. 1, and kept at nearly 30-70 and 90 °C, respectively. The
balloon and flask were filled with ultra-pure water. The round bottom borosilicate flask was bubbled continuously
for compensation to limit the air drying due to continuous injection of $N_2/O_2$. A stainless steel fan, mounted at the
bottom of the CESAM homogenized the aerosols and gas phase concentrations, temperature, and RH in the
chamber.

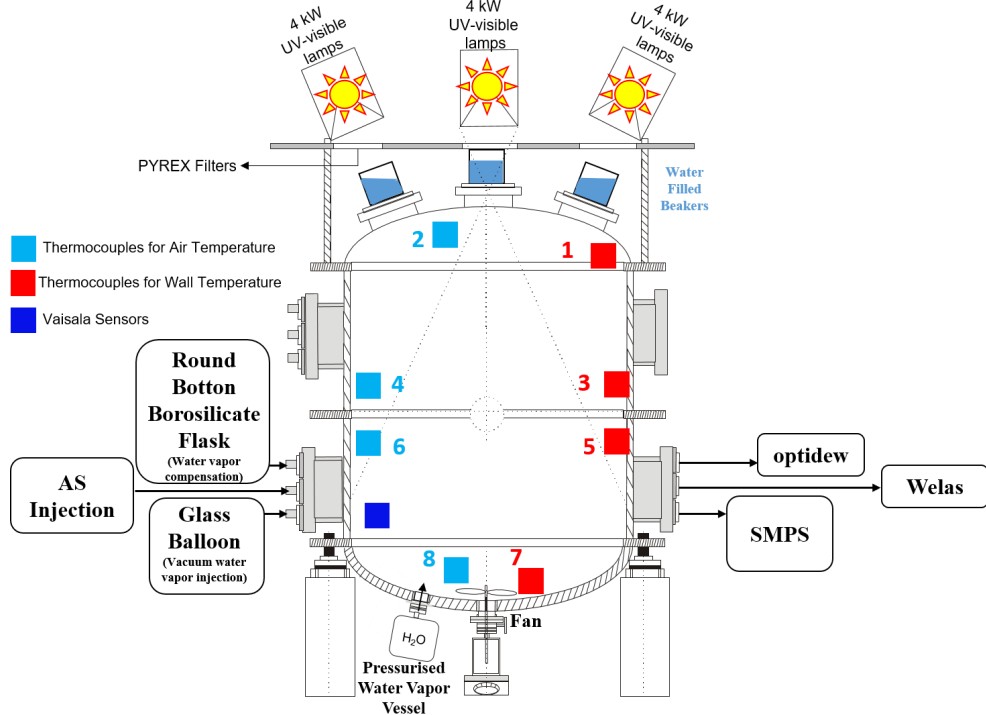


**Figure 1: Schematic front view of the CESAM with the positioning of various instruments and sensors (Wang et al., 2011).**


**2.1 Experimental protocol of liquid droplet cloud generation**


### 2.1.1 Cleaning Protocol

Previous studies have shown that cloud chamber experiments aiming at studying the aerosol-cloud processes are extremely sensitive to chamber cleanliness (Brégonzio-Rozier et al., 2015; Duplissy et al., 2010; Frey et al., 2018). To minimize contaminations, a cleaning protocol was established, which includes a manual cleaning with ethanol and ultra-pure water after each experiment to remove the particles and semi-volatile compounds which may have deposited on the chamber walls. Then, the walls were heated at 40 °C for several minutes and then CESAM was vacuumed in the range of $6 \times 10^{-4}$ hPa for a minimum of two hours (Brégonzio-Rozier et al., 2015). Finally, the chamber was cooled and kept under vacuum overnight to perform controlled cloud experiments the next day. Before starting each cloud experiments, the chamber was filled at atmospheric pressure, and aerosol number concentration was measured using SMPS (scanning mobility particle sizer, TSI 3080) to ensure that the particle number concentration was below 100 $cm^{-3}$, considering the chamber background level approximately 1-2% of the maximum seed particle concentration injected in the chamber so that effect of background could be negligible. If particle number concentration or mass were above the specified limit (100 $cm^{-3}$/$10^{-2}$ µg $m^{-3}$), the filling and flushing cycle of the chamber was performed again until these limits were achieved. This limit ensures the cleanliness of the chamber.

### 2.1.2 Cloud Generation Protocol

The cloud generation protocol was designed to investigate cloud microphysical properties under dark and light conditions. Using the expansion technique, several clouds could be generated during a single experiment. The optimized protocol was as follows: 1/ under vacuum ($10^{-4}$ hPa) the temperature was adjusted to 15-16 °C by regulating the coolant temperature, allowing the chamber temperature to be close to the surrounding laboratory one to maximize temperature (T) homogeneity within the whole chamber. 2/ Water vapor (nearly 61-63 g) was introduced under vacuum ($10^{-4}$ hPa) using a heated bulb connected to the chamber in order to reach nearly 85-95% relative humidity. Ultra-pure water from Fisher Scientific (LC-MS Grade) was used to limit impurities. 3/ The chamber was then filled with $N_2/O_2$ at 10 hPa above the ambient pressure (to avoid any contamination from the external air), analytical instruments were connected and started to sample, so water vapor "compensation" was switched on to limit the air drying. It should be noted here that RH was extremely difficult to maintain at high values (85-95%). As the chamber was filled, the temperature increased leading to a decrease of RH. So additional water had to be injected to reach again the target high RH values needed for cloud generation. To do so, the pressurized stainless-steel vessel was used as it allows increasing the RH by several % within a few seconds. 4/ Ammonium sulfate (AS) seed aerosol particle injection was started and stopped when desired seed concentration was achieved. The fan was switched on during particle injection. 5/ Prior each cloud run, the pressure was set to nearly 1090 mbar, and as soon as T reached stabilized values at RH > 90%, the chamber was rapidly pumped down to nearly 890 hPa (at 100 lpm). The pressure decrease leads to nearly adiabatic expansion, resulting in quasi-adiabatic cooling and the development of sufficient supersaturation to form cloud droplets. Seed particles activated and formed cloud droplets due to the achieved supersaturation, which is called the peak supersaturation. During the entire experiment, i.e. before, during and after the could event, the chamber wall temperature was continuously controlled and maintained above the dew point to avoid water condensation on the walls, which could occur accidently - in this case, no cloud was observed. Once the cloud event was over, the chamber was refilled with


N₂/O₂ for the next cloud generation. In a single experiment, between 1 to 4 clouds were generated using this
protocol.
All experiments were carried out with ammonium sulfate (AS) aerosol seed particles generated from a solution of
0.11 M ammonium sulfate solution with the highest possible purity (99.9999%, Merck) to avoid as much organic
contaminations (Wu et al., 2022). The solution was nebulized by atomization using a constant output atomizer
(TSI, model 3076) operated at a flow rate of 1.8 and 2.7 l m⁻³, respectively during light and dark experiments. A
Nafion™ dried the resulting droplets at RH below 25 % prior injection in the chamber. The target seed
concentration in the chamber was fixed at around 6000-8000 cm⁻³. An aerodynamic aerosol classifier (AAC,
Cambustion) was used to select monodisperse particles of 300 nm aerodynamic diameter (corresponding to ~219
nm mobility diameter assuming spherical particles of density 1.776 g cm⁻³).

### 176 2.1.3 Instrumentation

#### 177 2.1.3.1 Thermodynamic Measurements

Temperature and relative humidity (RH) were monitored using Vaisala® humidity and temperature sensors
(HMP234, Humicap®). The sensors measured the RH with an accuracy of 3% in pressurized and vacuum
conditions. One has to note that the time resolution of the temperature sensors was too low to allow for accurate
measurements during the fast chamber evacuation at 100 lpm, i.e. during most of each cloud event. As the cloud
formation was strongly depends on air temperature, four additional T-type thermocouples were installed at various
locations of the sensing volume of the chamber (see Fig. 1) to measure the air temperature variation before, during,
and after each cloud event, with an accuracy of ± 0.5 °C. The wall temperature was also monitored using four
additional T-type thermocouples to ensure that the wall temperature was above the dew point so to prevent
condensation. The top, middle-upper, middle-lower, and bottom wall temperatures were measured with $T_1$, $T_3$, $T_5$,
and $T_7$ sensors, respectively, whereas $T_2$, $T_4$, $T_6$, and $T_8$ sensors recorded the chamber's top, middle-upper, middle-
lower, and bottom air temperatures, respectively. All T sensors are installed so they do not have direct exposure
to incoming light to prevent from artificial heating. In addition, the thermocouples for the air temperature
measurements are fastened at an appropriate distance from the wall to avoid the influence of the wall temperature.
Additionally, a hygrometer (Chilled Mirror, Michell Optidew model 501) was henceforth connected to the
chamber to record the dew point temperature and the gas-phase water content, i.e. absolute humidity.

#### 193 2.1.3.2 Aerosol and Cloud Microphysical Properties

The size distribution of cloud droplets was continuously measured during the experiments with a time resolution
of 10-s using a white light optical particle counter (OPC) (Welas® 2000, Palas, flow rate: 2 l min⁻¹) (Brégonzio-
Rozier et al., 2015). It measured the cloud droplet's size distribution from 0.25 to 17.17 μm in optical size, using
the refractive index of water (1.33 ± 0i). It was calibrated by means of a calibration dust called CalDust 1100,
whose refractive index was (1.59 ± 0i). The Welas measured concentrations per size are corrected for sampling
losses in the tubes (von der Weiden et al., 2009), as well as for losses on the chamber walls and dilution (Wang
et al., 2011).
The AS seed particle size distribution was continuously recorded at 3-min time resolution using a Scanning
Mobility Particle Sizer (SMPS), consisting of a Differential Mobility Analyzer (DMA, TSI, model 3080) coupled



with a Condensation Particle Counter (CPC, TSI, model 3010). The instrument is operated at a flow rate of 1 l
$min^{-1}$ resulting in a nominal mobility size range of 19.5 – 881.7 nm. The SMPS was operated without dryer. The
sampling tube from the chamber to the SMPS was kept as short as possible, so that the measured size distribution
represented nearly the seed particle size distribution in the humidified chamber.
**3 Data Analysis**
**3.1 Cloud Formation Properties (CFPs)**
A significant part of our data analysis aimed at distinguishing between two (hydrated/inactivated particles and
cloud droplets) populations. In addition, the dry seed particle size distribution was not measured; therefore, due to
this limitation, it was necessary to retrieve the dry size distribution.
The Köhler theory (Köhler, 1936) considers that a seed aerosol particle becomes activated into a cloud droplet
when its dry or hydrated/wet size is similar to or larger than a threshold dry particle and droplet diameter,
respectively. These dry and wet diameters are respectively called critical dry diameters ($D_{crit}$) of a seed particle
and threshold droplet diameter ($D_{drop,thres}$). Characterizing these two parameters is the key to describe the
supersaturation state of the studied environment. To do so, various approximation techniques are reported in the
literature. Prabhakaran et al. (2020) and Shawon et al. (2021) reported $D_{drop,thres}$ as the separation diameter between
the inactivated/hydrated aerosol particles and cloud droplets in the cloud particle size distribution (measured by
Welas) as well as in the derived probability density function from the distribution. However, the lognormal size
distribution sometimes exhibits no distinct dip to characterize the threshold diameter. Instead, Hammer et al.
(2014) used the surface size distribution than the number size distribution to calculate $D_{drop,thres}$. Elias et al. (2015)
found that the inactivated/hydrated aerosol and fog droplets could be identified in the two modes of the volume
lognormal distribution aerosol particles measured by a Welas at ambient conditions, and defined $D_{drop,thres}$ as the
intersection/transition diameter between these two modes. However, in the present study, none of these approaches
lead to identifying a robust and stable dip in size/surface/volume distribution. To overcome this difficulty, an
alternative iterative approach, illustrated in Fig.2, was adopted to derive CFPs like cloud droplet concentration
($N_{drop}$), critical dry diameter of seed particle ($D_{crit}$), cloud droplet threshold diameter ($D_{drop,thres}$), and peak
supersaturation ratio ($s_{peak}$).





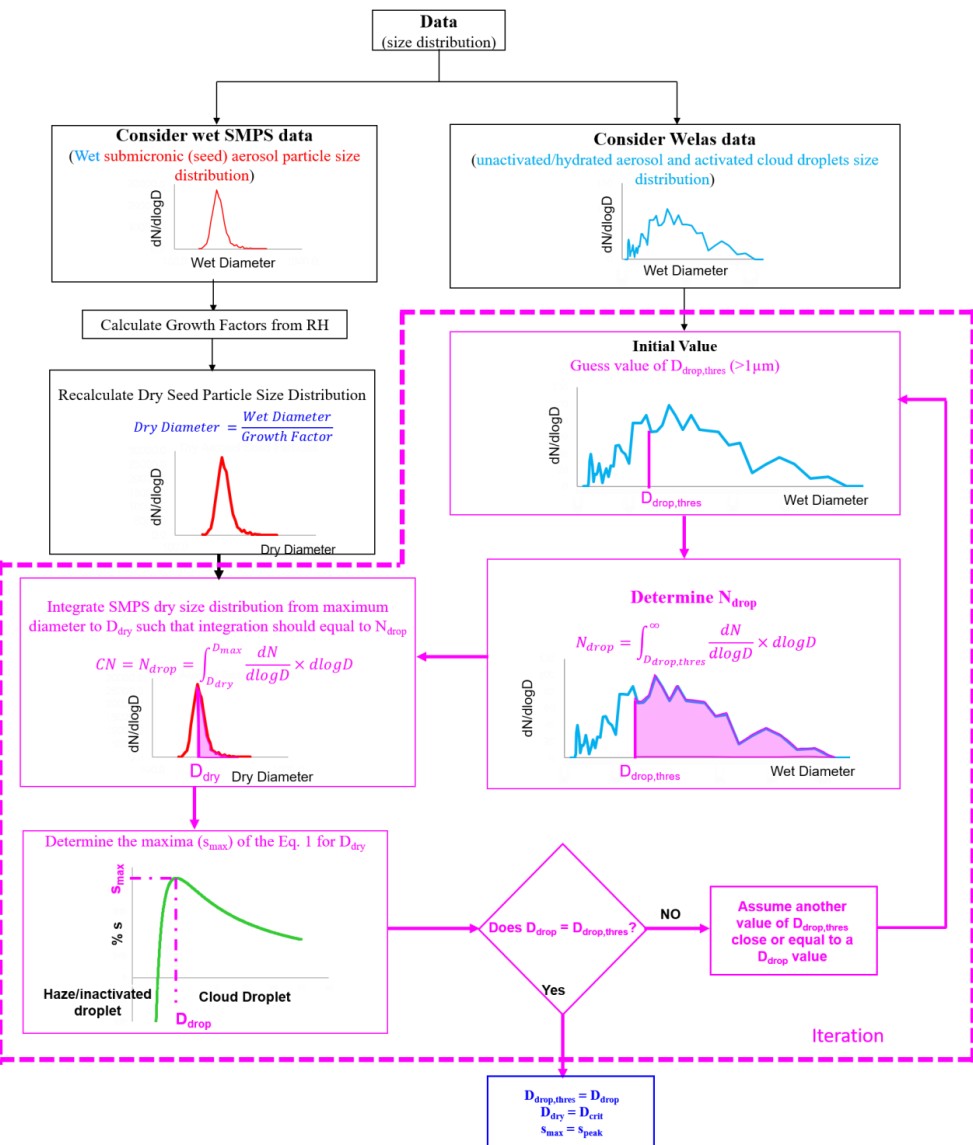


**Figure 2: Algorithm of the iterative scheme used to retrieve cloud formation properties (CFPs) from the wet particle size distribution.**




The model starts by considering the dry seed aerosol particle diameter ($D_{dry}$) and droplet diameter ($D_{drop}$), which
are linked to the peak supersaturation ratio (s) and aerosol hygroscopicity parameter ($\kappa$) through the $\kappa$-Köhler
equation (Petters and Kreidenweis, 2007) (Eq.1).
$$s = \frac{D_{drop}^3 - D_{dry}^3}{D_{drop}^3 - D_{dry}^3(1-\kappa)} \exp\left(\frac{4 \times \sigma \times M_w}{R \times T \times \rho_w \times D_{drop}}\right) - 1,$$
    (1)



Where σ is the surface tension at the droplet surface/air interface assumed equal to the surface tension of pure
water, R is the universal gas constant, $M_w$ is the molecular weight of water, T is the chamber air temperature in
Kelvin, $\rho_w$ is the density of water, and κ is the hygroscopicity parameter of the CCN (in our case, for AS, κ = 0.61.
It should be noted than our experiments did not benefit from CCN counter data hence the number concentration
of the CCN was not measured by set equal to the number of droplets ($N_{drop}$).
During our experiments it was observed that middle down ($T_6$) air temperature sensor was the most sensitive to
the T changes during the adiabatic expansions. Therefore, $T_6$ was used to determine the CFPs. The CFPs were
derived for each cloud as follows: $N_{drop}$ was set to a corresponding initial guess value of $D_{drop,thres}$ (> 1 µm), and
determined by integrating cloud droplet number concentration above $D_{drop,thres}$ using loss and dilution corrected
Welas measurements. The dry SMPS number size distribution was recalculated by the wet SMPS size distribution
considering the growth factor of AS particles. Hereafter, the measured dry SMPS size distribution represents the
retrieved one from the measured wet SMPS size distribution in the subsequent text. $D_{dry}$ was approximated by
integrating SMPS dry particle number size distribution from the maximum size ($D_{max}$) to a lower limit diameter at
which the estimated CCN matched the droplet number concentration ($N_{drop}$), as done by Lamb and Verlinde,
(2011), who calculated $N_{drop}$ according to Eq. (2).
$$N_{drop} = \int_{D_{dry}}^{D_{max}} \frac{dN}{dlogD} \times dlogD,$$     (2)
$D_{drop}$ was calculated by numerical searching the maximum of Eq.1 for $D_{drop}$. This threshold diameter of the cloud
droplet ($D_{drop,thres}$) define the size separation between the non-activated droplets. This parameter was assumed to
reach instantaneous equilibrium with the chamber effective supersaturation, and the activated cloud droplets, either
growing or shrinking in response to the chamber effective supersaturation to which they were exposed. The
resulting $D_{dry}$, called $D_{crit}$, indicates that the seed aerosol particles larger than $D_{crit}$ in size were activated into cloud
droplets equal or larger than $D_{rop,thres}$ in size.
The peak supersaturation ratio ($s_{peak}$) can be further determined by combining $D_{crit}$ and seed particle hygroscopicity
(for AS, κ = 0.61) using the κ-Köhler equation (Petters and Kreidenweis, 2007) as Eq. (3):
$$s_{peak} = \frac{2}{\kappa^{0.5}} \times \left( \frac{4 \times \sigma \times M_w}{3 \times R \times T \times \rho_w \times D_{crit}} \right)^{\frac{3}{2}},$$     (3)
The peak supersaturation can be described as a combination of the source and sink of the water vapour in the
chamber. $N_{drop}$ is the total cloud droplet concentration at effective supersaturation ratio, $s_{epeaak}$. Thus, iterations are
performed on $D_{drop,thres}$ until $D_{drop,thres}$ equals to $D_{drop}$. A solution to the iteration only exists in one trio point of $N_d$,
$D_{crit}$, and $D_{drop}$, which is related to $s_{peak}$. The iterations were performed for every time step of the expansion and
corresponding derived CFPs. Panel (c) in Fig. 3 shows the cloud droplet size distribution using this iterative
approach.
Furthermore, as a result of the iterative model, the particle activation ratio ($A_{cd}$) can be calculated using Eq. (4),
as performed by (Frey et al., 2018):
$$A_{cd} = \frac{N_{drop}}{N_s},$$     (4)



where $N_{drop}$ is the cloud droplet number concentration and $N_s$ is the pre-expansion aerosol seed particle number
concentration. Here, $N_s$ indicates the actual seed particle present in the CESAM for cloud formation, not corrected
for particle losses on the chamber walls and neither by dilution. The idea is that the $N_s$ indicates the actual number
of seed particles present in the CESAM available for cloud droplet formation processes.
**4 Results and Discussion**
The timeline of a typical cloud run is shown in Fig. 3. The chamber evacuation, indicated by the pressure drop in
the chamber, results in an adiabatic temperature drop in the air for the initial minute (Fig. 3a) but then, when the
liquid water content starts raising (Fig. 3d), the temperature decreases more slowly (Fig. 3a). This change in the
cooling rate can be due to the heat released by water condensation or/and by heat exchange with the chamber walls.
Initial fast cooling through adiabatic expansion creates the supersaturation ($s_{peak} \geq 0.035$ %; Fig. 3d) required to
activate the AS seed particles into cloud droplets. Then the fractional activation of seed particles into cloud droplets
leads to a mixture of non-activated particles and cloud droplets, as shown in Fig. 3b. It can be observed that the
wet seed particles exhibit a bimodal size distribution, with a mode around 3-4 μm and another one around 10-12
μm. The first mode suggests a mixture of hydrated but not fully activated particles and activated particles, i.e.,
droplets. Even if supposedly homogeneous physical conditions applied to a single aerosol distribution should lead
to a single droplet size distribution, it is an experimental fact that it is not the case here. Following the iterative
method mentioned in the previous section 3.1, the threshold droplet diameter that was determined and is shown in
panel (b) (solid red line). As a result, panel (c) shows the time series of the size distribution and total number of
activated cloud droplets.
Two types of experiments were conducted, one in presence of UV-vis irradiation, Irradiated Cloud (IC), and
another one without UV-vis irradiation, Non-Irradiated Cloud (N-IC). In IC and N-IC experiments, respectively 3
and 2 adiabatic expansions (cloud runs) were successively carried out, named IC-1, IC-2, and IC-3 for light
conditions and N-IC-1 and N-IC-2 for dark conditions (Table 1). Besides, N-IC-3* is marked with a star as it was
performed in a separate experiment. Following the cloud generation protocol mentioned in section 2.1.2, the cloud
lifetimes were found to range between $7.0 \pm 0.8$ and $6.3 \pm 1.4$ min in the presence and absence of light, respectively,
which is satisfying considering those in the atmosphere (2-30 min (Colvile et al., 1997)).
The key parameters for each cloud run are reported in Table 1. The initial RH was calculated considering chamber
air temperature ($T_6$) and absolute humidity measured by the Optidew. The chamber evacuation rate (100 lpm) was
the same for all cloud runs to avoid any impact of the cooling rate on the CFP. In addition, Δp represents the net
pressure drop during the chamber evacuation, and ΔT indicates the net temperature drop as a result of quasi-
adiabatic expansion. It is interesting to note that ΔT increased in successive cloud runs in each experiment (e.g.
N-IC-1 to N-IC-2 and IC-1 to IC-2 and IC-2 to IC-3). The duration of the pressure drop was manually controlled,
thus explaining the different ΔP values shown in Table 1. Notably, the initial temperature for a cloud run increased
in the successive cloud runs during an experiment, especially during IC experiments, due to the heat generated by
the lamps, despite the IR filtering. Considering the net temperature drop during the expansion, the mean cooling
rate for N-IC-1 (0.36 °C/min) was found to be comparable to that for IC-1 (0.33 °C/min), while the 0.42 °C/min
cooling rate for IC-2 and IC-3 was nearly similar to N-IC-2 (0.43 °C/min).



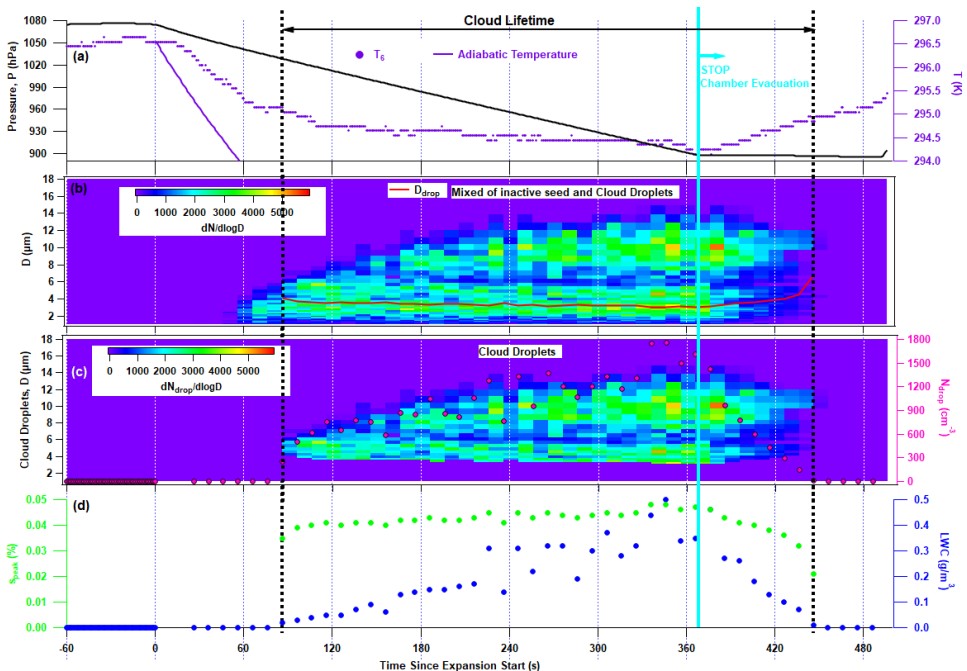


**Figure 3: Example of a cloud formation process on 215 nm ammonium sulfate seed particles. Panel (a) shows the pressure drop and a nearly adiabatic temperature drop during the initial minute following slow temperature drop, (b) size distribution of hydrated/non-activated seed particles and droplets, (c) size distribution and total number concentration of droplets, (d) peak supersaturation ($s_{peak}$) and cloud liquid water content (LWC) versus time.**


**Table 1. Initial parameters for all cloud runs** – IC and N-IC denote cloud runs performed under Light and Dark conditions
respectively. The numbers correspond to successive expansions within 1 experiment, except for N-IC-3* which was performed
in a separate experiment. ** indicate the RH value measured using Vaisala sensor. The error indicates the measurement error.

| Parameter | Cloud Run | | | | | |
|---|---|---|---|---|---|---|
| | N-IC-1 | N-IC-2 | N-IC-3* | IC-1 | IC-2 | IC-3 |
| p (hPa) | 1078.6 | 1078.0 | 1150.0 | 1074.7 | 1070.9 | 1099.3 |
| T (°C) (±error) | 17.5 ± 0.5 | 18.0 ± 0.5 | 16.4 ± 0.5 | 23.4 ± 0.5 | 24.0 ± 0.5 | 25.0 ± 0.5 |
| RH (%) (±error) | 102.4 ± 2.9 | 98.4 ± 0.8 | 86.5** | 93.7 ± 0.7 | 93.4 ± 0.7 | 94.1 ± 0.7 |
| Air Evacuation Rate (lpm) | 100 | 100 | 100 | 100 | 100 | 100 |
| Air Evacuation Duration (min) | 6.13 | 8.07 | 7.18 | 7.2 | 6.97 | 6.57 |
| Dew Point (°C) (±error) | 17.9 ± 0.15 | 17.7 ± 0.15 | | 22.3 ± 0.15 | 22.9 ± 0.15 | 24 ± 0.15 |
| Δp (hPa) | -199.3 | -198.6 | -199.4 | -176.9 | -230.2 | -211.6 |
| ΔT (°C) | -2.2 | -3.5 | -1.1 | -2.4 | -2.9 | -3.4 |


### 4.1 Cloud Runs without UV-vis Irradiation

The cloud formation properties (CFPs) and the timeline of the non-irradiated cloud run N-IC-1 are illustrated in
Fig. 4 (N-IC-2 and N-IC-3* cloud runs are illustrated in the Supplementary material). The initial time (0 s) in the
panels indicates the starting time of the expansion. The upper panel (Fig. 4(a)) shows the dry seed particle size
distribution, measured by the SMPS before and after chamber evacuation. The total seed particle concentration
was 5240 cm$^{-3}$ just before and 3683 cm$^{-3}$ after the chamber evacuation. The difference, of the order of 30%, is in
agreement with the loss of seed particles due to the combination of the dilution in the chamber during its evacuation
(nearly 12 %), the wall losses and the sedimentation losses of cloud droplets, jointly accounting for the remaining
18%. The lifetime of particles in the 200 nm diameter range is in the order of 2 to 3 days in the CESAM chamber,
but of a few minutes only for droplets of several micrometres in diameter (see Fig. S1 in Lamkaddam, 2017). For
comparison, Chang et al. (2016) reported that approximately 83% of seed particles may be lost due to cloud
droplets losses, excluding dilution. This much larger loss can be explained by two factors; first, in Chang et al.
(2016) the droplets were larger (approximately 5-20 μm due to higher supersaturation and more hygroscopic seed
particles (NaCl)) and second, their chamber was smaller with a different shape, inducing larger wall losses.
The other panels in Fig. 4 exhibit the time series of the parameters during cloud development. The cloud droplet
growth starts from at 42 s onwards, evidenced by the significant enhancement of all the parameters shown in Fig.
4c, 4d, and 4e. The time elapsed to achieve the $s_{peak}$ value required to start seed particle activation highly depends
upon the initial chamber conditions before the evacuation (Frey et al., 2018) as well as upon the rate of evacuation
and thus the cooling rate (that was fixed to 100 lpm in our case). This elapsed time cannot be attributed to the
transit time through the sampling tube from the chamber to the Welas instrument as it is lower than one second in
our set up. In agreement with Möhler et al. (2003) we rather explain this to chamber boundary layer effects as the
chamber air near the walls could remain cloud-free because the wall temperature is maintained slightly higher than
inner air. This delay between the start of the evacuation and the first droplets detection was observed for all the
cloud runs, in the range from 50 to 110 s, in agreement with similar studies (Frey et al., 2018).
An initial high concentration of tiny cloud droplets (1.9-8.8 μm) is observed (orange to reddish colors in Fig. 4(c))
when approximately all seed particles are activated into droplets. These tiny droplets then grow as the cloud run
proceeds while the small droplet mode vanishes. The initial high RH (99%; Table 1) is probably responsible for
reaching a sufficient supersaturation (> 0.078%) in the chamber to activate all the seed particles into droplets after
50 s initiation time. The chamber peak supersaturation varied from 0.025 to 0.079% (Fig. 4(d)), however, the
maximum $s_{peak}$ could be even higher than 0.079% in the chamber because this value is the limit constrained by our
method to calculate $s_{peak}$ as cloud droplets could not be more than seed particles. Any fluctuations in the $s_{peak}$ values
could be explained by chamber turbulences (Prabhakaran et al., 2020) which proportionately also impacts the
cloud evolution (e.g. LWC values fluctuations in Fig. 4c).
The cloud droplets' volume mean diameter (MVD) ranged from 6.1 μm (at the initial stage) to 11.8 μm, with a
mean value of 9.4 ± 2.0 μm (Table 2), which is consistent with the values reported by Frey et al. (2018) for liquid
clouds droplets formed on AS seed particles. As smaller droplets grow into larger ones or coagulate, they make up
the LWC that reaches a maximum of 1.8 g m$^{-3}$ (Fig. 4(d)) after 250 s, when a significant fraction of the formed
droplets grow to bigger droplets which correlates well with the maximum MVD. However, the mean LWC was
1.0 ± 0.4 g cm$^{-3}$, significantly higher than 0.5 g m$^{-3}$ (maximum) reported by Frey et al. (2018) for non-irradiated
cloud. The significantly higher LWC in the present study was found due to the higher seed and droplet
concentration.
The detailed values of all cloud formation parameters are mentioned in Table 2. For N-IC-1, the mean (± std)
values for particle activation ratio ($A_{cd}$) was observed at 0.98 ± 0.37. Fig. 5e shows that $A_{cd}$ values were frequently
higher than 1, which could be due to the instrument margin errors (Frey et al., 2018). Finally, cloud N-IC-1





sustained for 6 min 50 s (thus 78 s after the stop of the chamber evacuation), well within a typical atmospheric
cloud droplet lifetime of 2-30 min (Colvile et al., 1997; Herrmann, 2003).
Assuming that successive cloud runs did not impact the chemical composition of seed particles, we compare in
the following all the N-IC cloud runs as independent runs and/or experiments. The microphysical properties of N-
IC-2 and N-IC-3* are illustrated in Fig. S2 and S3, respectively. N-IC-2 displayed a 6 min lifetime, shorter than
N-IC-1 (7 min 50 s), while N-IC-2 showed significantly ($p<0.05$) lower $A_{cd}$ and MVD than N-IC-1 (Table 2). This
could be due to the significantly ($p<0.05$) lower peak supersaturation which was always $\leq 0.050\%$ in N-IC-2 (Fig.
S2), while it was probably often higher than the maximum value (0.079%) in N-IC-1. It is worth noting that the
initial temperature and RH values for N-IC-1 are respectively slightly lower and higher than in N-IC-2, as shown
in Table 1. This could support the higher $s_{peak}$ achieved in N-IC-1. In addition, the pressure drop ($\Delta p$) was nearly
the same for both clouds, while cooling ($\Delta T$) was higher for N-IC-2. It indicates that even high cooling in N-IC-2
could not generate a sufficiently high degree of supersaturation that could lead to a high activation ratio. It is thus
likely that the observed different supersaturations were mainly due to the initial RH conditions. In addition, N-IC-
3* (Fig. S3), with a lower seed concentration, shows lower cloud droplet concentrations even though the
supersaturation ratio is comparable to N-IC-2, negatively impacting LWC. Notably, the mode of dry seed size
distribution in N-IC-3* was 17 nm lower than that of N-IC-2, which could be the reason for lower $A_{cd}$ in N-IC-3*
than in N-IC-2 at nearly the same $s_{peak}$, because lower size particles require higher supersaturation to activate into
droplets (Köhler, 1936). In N-IC-3*, Fig. S3 shows that $N_{drop}$ was extremely low and variable, as well LWC,
and thus it is difficult to provide any information on the droplet's growth. All these observations explain why N-
IC-3* showed a significantly lower MVD ($p<0.05$) than the two other dark clouds.










**Table 2: Detailed values of microphysical parameters of dark clouds (N-IC) and light clouds (IC).**

| Cloud Run | Cloud Life min | $N_s$ cm$^{-3}$ | Dry seed Model Diameter nm | Critical Droplet Diameter Minimum µm | $N_{drop}$ cm$^{-3}$ | Mean Volume µm | $A_{ct}$ | $S_{peak}$ % | LWC g/m$^3$ |
|---|---|---|---|---|---|---|---|---|---|
| | | | | | | Mean ± STD (max) | | | |
| **N-IC-1** | 7.83 | 5240 | 224 | 1.9 | 5171 ± 1946 (8725) | 9.4 ± 2.0 (11.8) | 0.98 ± 0.37 (1.65) | 0.07 ± 0.02 (0.08) | 1.0 ± 0.4 (1.8) |
| **N-IC-2** | 6 | 3672 | 218.7 | 2.9 | 1153 ± 629 (2176) | 8.0 ± 1.3 (9.6) | 0.31 ± 0.18 (0.59) | 0.04 ± 0.01 (0.05) | 0.17 ± 0.10 (0.32) |
| **N-IC-3\*** | 5.17 | 1927 | 201.7 | 3 | 240 ± 186 (658) | 5.7 ± 1.2 (7.4) | 0.13 ± 0.09 (0.34) | 0.04 ± 0.01 (0.05) | 0.02 ± 0.02 (0.05) |
| **IC-1** | 6.2 | 2987 | 215 | 3 | 935 ± 432 (1757) | 8.5 ± 1.6 (10.4) | 0.31 ± 0.14 (0.58) | 0.04 ± 0.00 (0.05) | 0.20 ± 0.13 (0.50) |
| **IC-2** | 7.83 | 1942 | 211.7 | 1.8 | 1215 ± 555 (2230) | 9.6 ± 1.6 (11.4) | 0.62 ± 0.28 (1.14) | 0.05 ± 0.01 (0.08) | 0.37 ± 0.22 (0.76) |
| **IC-3** | 7 | 1158 | 216 | 2 | 689 ± 484 (1814) | 10.4 ± 1.9 (13.6) | 0.59 ± 0.41 (1.55) | 0.05 ± 0.01 (0.07) | 0.32 ± 0.31 (1.35) |

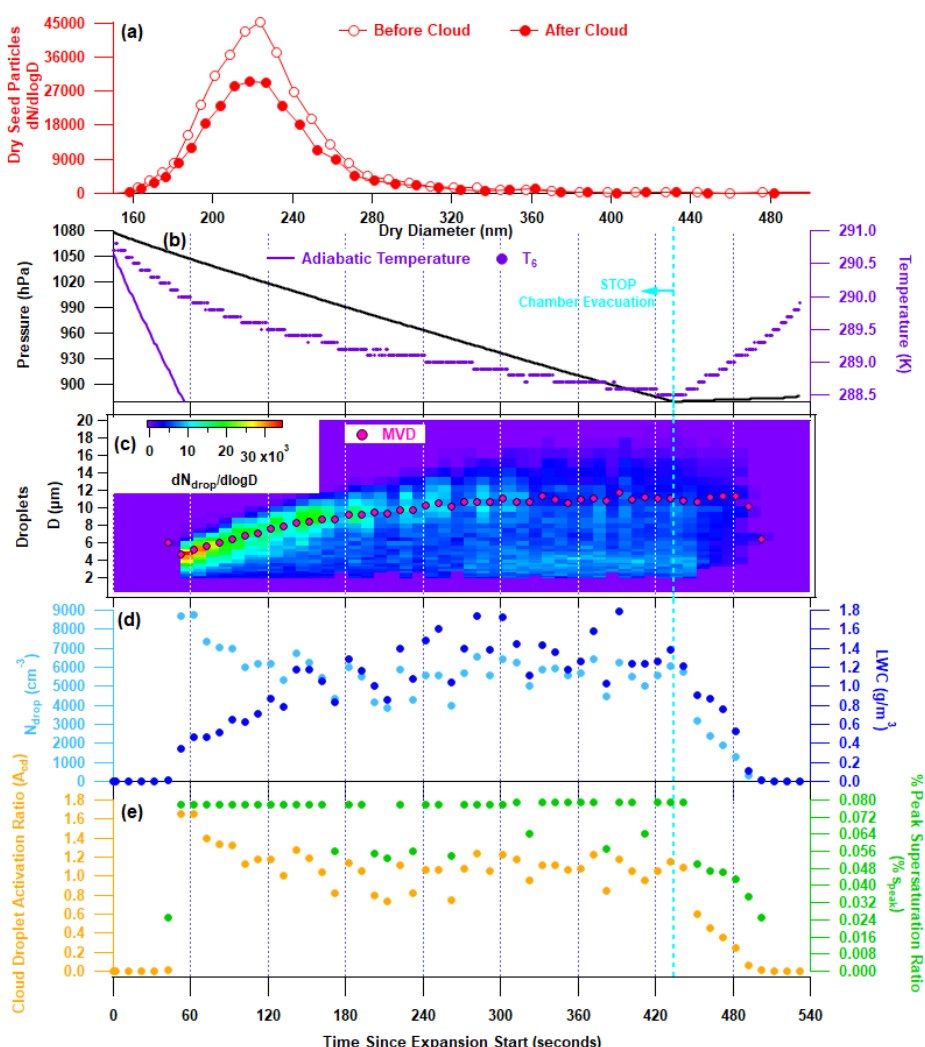


**Figure 4: Example of a cloud run (N-IC-1) performed without UV-vis irradiation using 224 nm ammonium sulfate seed particles. Panel (a) shows the SMPS size distributions of seed particles obtained before and after the cloud run, (b) shows the pressure drop and a nearly adiabatic temperature drop during the initial minute, following slow temperature drop, (c) time series of cloud droplet size distribution and volume mean diameter (MVD) measured by WELAS, (d) time series of cloud droplet concentration ($N_{drop}$) and LWC, and (e) time series of seed particle activation ratio ($A_{cd}$) and chamber peak supersaturation ratio ($s_{peak}$).**

### 4.2 Cloud Runs under UV-vis Irradiation

Fig. 5 illustrates the IC-1 cloud run. IC-1 was carried out by reducing the pressure by 176.9 hPa from 1074.7 hPa, leading to a temperature drop of 2.4 °C. The initial temperature and RH were 23.4 °C and 93.7 %, respectively (Table 1). The temperature decay at the beginning of the expansion is nearly parallel to the adiabatic temperature. The total number concentration of seed aerosol particles was 2987 cm$^{-3}$ before the cloud run, significantly



decreasing to nearly 1942 cm$^{-3}$ (35% loss) due to seed particle loss and dilution. The dilution would only have
reduced the concentration to 2615 cm$^{-3}$ (12 %) the remaining seed particles must have been removed from the
chamber due to cloud droplet sedimentation and/or wall loss. The critical dry diameter of the seed particles reached
nearly 219 nm when maximum 58% seed particle were activated into droplets, as shown in panel 5(a), leading to
$s_{peak}$ values reaching upto 0.048% during the cloud formation process. The value of 219 nm is in the lower range
of the number-size distribution measured after the chamber's evacuation and re-pressurization. This suggests that
some seed particles activated into cloud droplets while smaller ones did not. Chang et al. (2016) observed a similar
feature in their experiments with NaCl seed particles.
The droplet number size distribution during the cloud formation is shown in Fig. 5c. No cloud droplets are observed
until 85 s after expansion, when a few seed particles (253 cm$^{-3}$) activated to droplets at a size range of 3.9-4.2 μm.
Afterwards, the cloud development accelerates, and small droplets are formed. As the cloud run proceeds, the
droplet size distribution shifts to a larger size, while the small droplet mode slightly decreases (Fig. 5c). After the
first 4 min, the MVD stabilises at 8.2 ± 1.6 μm. The growth of the large droplets is well reflected by the increasing
LWC, which maximum value is consistent with the approximate value of 0.5 g m$^{-3}$ reported by Frey et al. (2018)
although their study was conducted in the dark. The peak supersaturation ($s_{peak}$) reached 0.048% when 58% of
seed particles were activated into cloud droplets. The $s_{peak}$ builds up as the cloud run proceeds and reaches its
maximum value just before the end. The fluctuations in $s_{peak}$ value can be explained by the inhomogeneous RH
and temperature profiles in the chamber. Despite continuous mixing, the chamber walls temperature is kept at
controlled values while the center of the chamber cools quasi-adiabatically during a chamber air evacuation. This
can develop a temperature gradient inside the chamber, a high temperature to lower temperature at the centre of
the chamber, causing a humidity gradient (Hinds, 1999), which impacts cloud microphysics. In addition, the
evacuation creates turbulences in the chamber (as mentioned in the previous section) and causes a mixing inside
the chamber, creating inhomogeneities in the temperature and RH profiles. This turbulence also leads to a
broadening of droplets distribution, $A_{cd}$, and $s_{peak}$ (Abade et al., 2018; Prabhakaran et al., 2020). The cloud droplet
activation ratio, $A_{cd}$, varied from 0 to 0.58. Further, cloud IC-1 persisted for nearly 5 min, which is shorter than
dark cloud N-IC-1. As they were performed under very similar initial T and RH conditions (Table 1), both IC-2
and IC-3 cloud runs (Figures S4 and S5) showed no significant differences in the mean value of particle activation
ratios, peak supersaturation ratios, and liquid water contents.












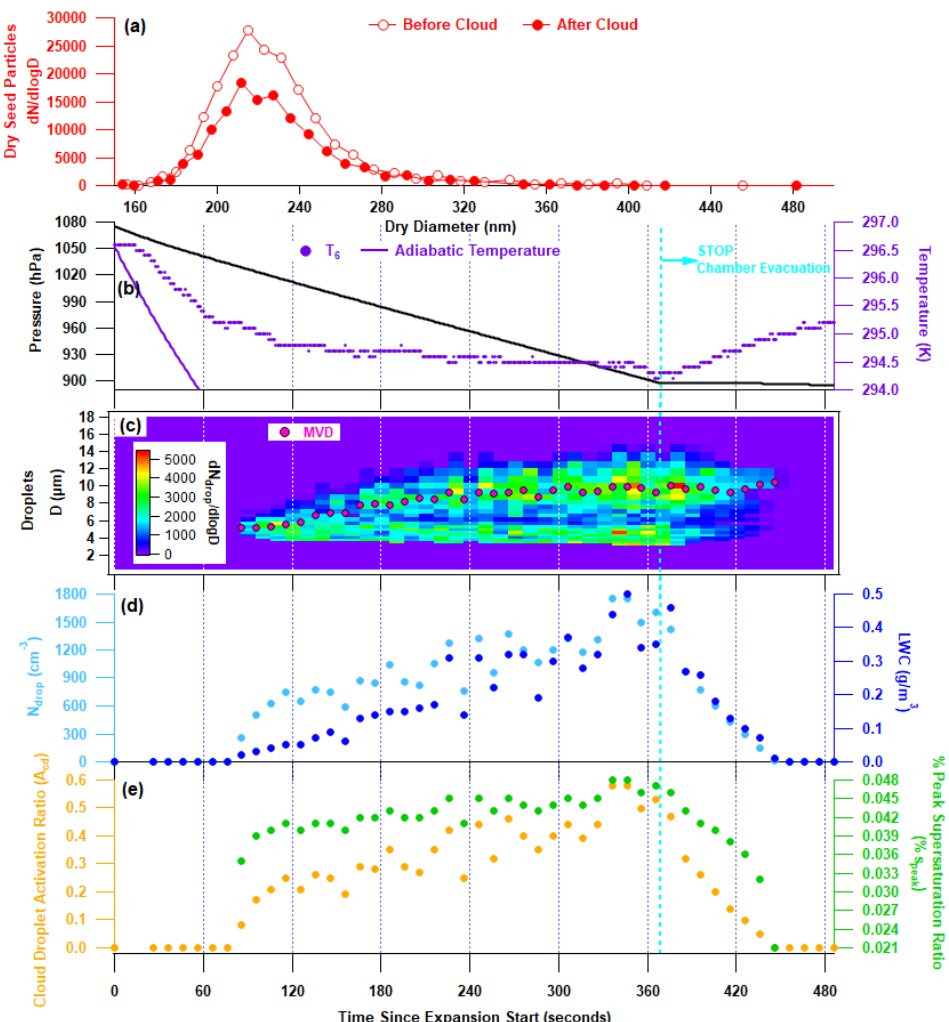

**Figure 5: Example of a cloud run (IC-1) performed under UV-vis irradiation using 215 nm ammonium sulfate seed particles. Panel (a) shows the SMPS size distributions of seed particles obtained before and after the cloud run, (b) shows the pressure drop and a nearly adiabatic temperature drop during the initial minute, following slow temperature drop, (c) time series of cloud droplet size distribution and volume mean diameter (MVD) measured by WELAS, (d) time series of cloud droplet concentration ($N_{drop}$) and LWC, and (e) time series of seed particle activation ratio ($A_{cd}$) and chamber peak supersaturation ratio ($s_{peak}$).**

### 4.3 Comparison between with and without Irradiated Clouds

Interestingly, there is a significant difference in the formation pattern between non-irradiated clouds (N-IC) and irradiated clouds (IC) (Fig. 4 and 5). The N-IC clouds exhibit a flash activation of all or maximum fractional seed particles into cloud droplets in the first minute following the start of expansion. Then a decrease followed by a nearly stable trend are observed in the cloud droplets and other microphysical properties. This flash activation



correlates well with the build-up of high supersaturation ratio ($s_{peak}$) at the initial stage. On the contrary, during
irradiated clouds (IC), microphysical parameters increase steadily during the expansion and reach their maximum
values at the last stage of each cloud run. This is the case for the supersaturation ratio $s_{peak}$ and the cloud droplet
activation ratio $A_{cd}$. Consequently, the LWC and cloud droplet number increased simultaneously as the cloud run
proceeds under irradiation conditions. Conversely, during N-IC, after the initial flash activation of all seed
particles, $N_{drop}$ decreases while the LWC increases (Fig. 4d and S2d), indicating that cloud droplets are coagulating
into bigger droplets after the initial activation. At a later stage (i.e. after ~ 150 s in Fig. 5b, when $N_{drop}$ remains
roughly constant while LWC still increases), these droplets grow due to water vapor condensation. However, under
irradiation conditions, activated droplets grow due to condensation throughout the cloud run without any
coagulation of smaller size droplets to bigger ones, as shown by the continuous increase of $N_{drop}$ and LWC (Fig.
5d, S4d and S5d). One of the possible reasons for these activation patterns could be that the heating of the chamber
air by the lamps counteracts the adiabatic cooling, leading to a reduced supersaturation at the beginning of the
cloud event, causing less droplets and less droplet growth. However, this direct heating effect is unlikely for two
reasons. First, the temperature measurements contradict this hypothesis (Fig. 5), as the temperature decay at the
beginning of the expansion is nearly parallel to the adiabatic temperature and show no significant difference with
the temperature decays recorded during the experiments without irradiation; Second, air is not and efficient light
absorber and, in our case, only water vapour could play the role of a greenhouse gas. Nevertheless, the light used
to irradiate the chamber was filtered by ca. 25 cm of liquid water removing the largest part of potentially warming
infrared radiation.
A second potential explanation is that ammonium sulphate enriched deliquescent particles absorb the remaining
(non-filtered) infrared radiation of the incoming light that warms them up, resulting in higher temperature than in
surrounding air. This causes a longer duration to stabilize the equilibrium between the droplet and the surrounding
atmosphere. This droplets/haze particle heating restricts the initial flash activation of seed particles and also
restricts the supersaturation at the initial stage of the cloud run. Nevertheless, when some droplets are formed, after
some time, the light seems to lose its importance so that the supersaturation and, thus, all related cloud parameters
could continuously increase with time. The incoming radiation is probably more reflected by the droplets, so the
interstitial chamber air could further cool to create a higher supersaturation.
**5 Conclusions**
The control of cloud formation under dark and light conditions is a prerequisite for further multiphase
photochemical studies in chambers under realistic conditions. This work aimed at optimizing a controlled protocol
for generating quasi-adiabatic expansion clouds of liquid droplets under atmospherically relevant simulated non-
irradiated (dark) and irradiated (light) conditions in the CESAM chamber. Successful experiments provided the
formation of 1 to 3 successive clouds within a single experiment, using an optimized protocol employing
monodisperse ammonium sulfate seed particles under dark and light conditions. This firmly demonstrates that,
although extremely challenging, especially under light conditions, it is possible to perform cloud experiments
under reproducible conditions in the CESAM chamber.
The expansion liquid clouds were a mixture of inactivated deliquescent seed particles and droplets. To discriminate
between them, an iterative approach was proposed to filter the cloud droplets from the mixture of hydrated seed





particles and droplets without any CCN counter instrument. The method allowed to determine microphysical
parameters, i.e. critical dry activation diameter of seed particle, threshold droplet diameter, peak supersaturation
ratio, number of cloud droplets, and seed particle activation ratio. The cloud lifetimes were found to be $7.0 \pm 0.8$
and $6.3 \pm 1.4$ min in the presence and absence of light, respectively. It falls, in the range of the lifetime of
atmospheric droplets (2-30 min). Some of the successive clouds within a single experiment showed very similar
properties.
The characterization of the formed liquid clouds showed specific trends in the microphysics parameters. Notably,
the seed particle loss at the end of the cloud was found to be a function of the fractional contribution of the largest
droplets due to their lifetimes in the CESAM chamber. Moreover, the cloud's liquid water content (LWC) was well
associated with the number of grown/larger size droplets. In addition, the achieved supersaturation was observed
as a function of initial chamber air temperature and relative humidity.
Non-irradiated cloud (N-IC) witnessed the activation of all seeds or a maximum of seed particles into droplets
within the first 1-2 min of the cloud, while Irradiated cloud (IC) took longer to activate all or part of the seed
particles into droplets. While still hypothetical, we explain this difference in the formation patterns with the
absorption of infrared light by the hydrated seed particles, inducing steep temperature gradients between each
hydrated particle and its surrounding environment. This indirect warming effect leads to a longer stabilization of
the equilibrium between the droplet and the surrounding atmosphere. This heating of the hydrated particles particle
restricted the flash activation of all seed particles and the higher supersaturation at the initial stage of the IC cloud.
Overall, at the later stages, the light intensity inside the cloud decreases so that the supersaturation and, thus, all
related cloud parameters could continuously increase with time. The light reflections by the droplets may increase,
causing the interstitial air cooling and thus higher supersaturation. This phenomenon should also impact the
air/water partitioning of semi-volatile compounds, and it should be considered for further studies, especially in
further multiphase photochemical studies implying water soluble volatile organic compounds in the CESAM
chamber.
**Data availability.** The data are available through the database of Atmospheric Simulation Chambers Studies
(DASCS) of the Eurochamp database hosted by the ACTRIS data center under the DOI xxxx findable the link
xxxxx (link under construction- will be provided before publication).
**Author contributions. AKM:** conceptualization, perform experiments, data analysis, investigation, methodology,
writing (original draft and review and editing). **JW:** conceptualization, perform experiments, methodology, writing
(review and editing). **AM:** experiment design, conceptualization, perform experiments, methodology, funding
acquisition, supervision, writing (review and editing). **PF:** experiment design, conceptualization, perform
experiments, data analysis, methodology, funding acquisition, project administration, supervision, writing (review
and editing). **BPV:** experiment designing, conceptualization, performing experiments, methodology, writing
(review and editing). **MC:** experiment designing, conceptualization, performing experiments, methodology,
writing (review and editing). **SM:** experiment designing, conceptualization, performing experiments,
methodology, writing (review and editing). **LP:** experiment designing, conceptualization, performing experiments,
methodology, writing (review and editing). **AB:** conceptualization, performing experiments, methodology, writing



(review and editing). **EP:** conceptualization, performing experiments, methodology, writing (review and editing).
**AT:** conceptualization, performing experiments, methodology, writing (review and editing). **TS:**
conceptualization, performing experiments, methodology, writing (review and editing). **LW:** conceptualization,
performing experiments, methodology, writing (review and editing). **HH:** experiment design, conceptualization,
methodology, funding acquisition, supervision, writing (review and editing). **JFD:** experiment design,
conceptualization, perform experiments, data analysis, methodology, funding acquisition, project administration,
supervision, writing (review and editing).
**Competing interest.** At least one of the (co-)authors is a member of the editorial board of Atmospheric
Measurement Techniques.
**Acknowledgements.** The AERIS data center (www.aeris–data.fr) for distributing and curing the data produced by
the CESAM chamber through the hosting of the EUROCHAMP datacenter (https://data.eurochamp.org). The
authors are also thankful to Mr. Nicolas Brun and Brice Temime-Roussel, Aix Marseille Université, CNRS, LCE,
Marseille, France and Majda Mekic, Leibniz Institute for Tropospheric Research, Leipzig (TROPOS), 04318,
Germany for helping in the experiments in the CESAM multiphase simulation chamber.
**Financial Support.** This work has received funding from the French National Research Agency (ANR) and the
German research Fundation (DFG) through the bilateral research project PARAMOUNT under the grant number
ANR-18-CE92-0038.It has received support from the European Union's Horizon 2020 research and innovation
program through the EUROCHAMP-2020 Infrastructure Activity under grant agreement no. 730997. CNRS-
INSU is gratefully acknowledged for supporting the CESAM chamber as a national facility as part of the French
ACTRIS Research Infrastructure.

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
