# Peer review of "Significant Influence of UV-vis Irradiation on Cloud Activation Efficiencies of Ammonium Sulfate Aerosols under Simulated Chamber Conditions"

_Atmospheric Measurement Techniques, 2023_

## Referee Comment (RC1)

Review of
**Significant Influence of UV-vis Irradiation on Cloud Activation Efficiencies of Ammonium Sulfate Aerosols under Simulated Chamber Conditions**
Author(s): Anil Kumar Mandariya et al. MS No.: amt-2023-206

**General comments**

The manuscript is not well written and it doesn't provide sufficient evidence for an influence of UV-VIS radiation on cloud activation nor for the suitability of the CESAM chamber for conducting cloud activation experiments in the set up presented. A rigorous assessment of the uncertainties e.g. resulting from sampling, temperature, and humidity measurements at different locations and the influence of gradients is most critical but not presented. Therefore, I can't recommend publication of this manuscript in Atmospheric Measurement Techniques.

**Specific comments**

Actually, the authors conducted only three independent experiments of which two were mainly compared (N-IC1 and IC1). For these two, the initial particle number concentrations (5240 vs. 2897 cm-3) and relative humidities (102.4±2.9% vs. 93.7±0.7%) were significantly different. The cloud activation observed was dominated by the initial relative humidities (RH) at a fixed and limited pumping speed. The authors admit this e.g. in line 379. Therefore, it is most likely that the differences observed for the experiments with and without illumination are caused by the generally lower initial RH during the experiments with light. The subsequent cloud activations with the remaining seed particles don't show a significant influence of a loss of the most active CCN in the previous activation(s) maybe due to the same limitation. The 'dry' seed particle size distributions before and after cloud processing also don't show a significant loss of the larger particle fraction.

Figures 3 and 5 show the same experiment with mainly identical information and may be merged. The authors claim that the dry seed particle size distributions given e.g. in figure 5 could be calculated from the SMPS data with the assumption that the SMPS measured the 'nearly' wet particle distribution (line 206). This is possible for a modified SMPS e.g. with humid sheath air but it is not clear if this was used.

The authors speculate that radiative heating of the seed aerosol particles leads to a different cloud activation behavior with and without irradiation. To substantiate this, a potential warming of the aerosol particles should at least be estimated/calculated based on the potential irradiation spectrum and its intensity. However, the irradiation spectrum isn't given especially in the infrared (Wang et al., 2011). Did you measure the impact of the water filters for the irradiation spectrum? For the statement that inside the clouds the light intensity deceases no evidence is presented (line 529).

The authors state that turbulences, inhomogeneities in temperatures and humidity affect the cloud activation (line 354 or 444) but don't give any estimation of turbulences, gradients of temperatures, or mixing time scales for their chamber. This is in strong contradiction to the aim of the manuscript to provide an optimized protocol to generate expansion type liquid clouds. It would have been helpful to compare the different temperature sensors for the gas and wall temperatures as well as the two hygrometers. Furthermore, it should be discussed if condensation of water on the temperature sensors could cause a positive bias for the gas temperature determination (line 283). Comparing the all sensor data with e.g. a fluid dynamic model run for an expansion experiment could help to understand the dynamics and e.g. how the heat fluxes from the chamber walls limit cloud homogeneity and lifetime.

A particle background of 100 cm$^{-3}$ (line 138) is a relatively high value which can have a substantial impact on cloud activation experiments depending on the type of particles. Why is this background so high and what is known about the nature of these background particles?

**Technical corrections and short comments**

Abstract: Please correct the English.
Line 34: affect
Line 35: of clouds
Line 39: droplets
Lines 51-53: Rephrase sentence
Line 55: What are inconvenient locations?
Line 61: the DRI chamber
Line 62: The reference Alpert et al. 2023 is missing in the reference list. Provide a correct reference to the handbook.
Line 65: processes
Line 68: What are 'long' liquid clouds? Add a reference on atmospheric cloud lifetimes.
Line 72: corresponding to an adiabatic ascent rate
Line 74: What are ammonium seed particles?
Lines 81-83: Rephrase and avoid simplification
Lines 86-87: Check for further studies like Vallon et al., AMT, 2022 (see figure S11 therein).
Line 88: Name the challenges
Line 93: give a reference to the PARAMOUNT project
Line 97: Why do the lamps have 3x6500 W but in figure 1 3x4000 W?
Line 113: consisted of a
Line 115: correct: high flow mass controller and give its range.
Line 117-118: Explain how the processes of injection and compensation work and are controlled as they seem to have substantial impact on generating relative humidities >90% in CESAM.
Line 121: Give more details on the fan (size, speed, mixing time scale) as these details seem to relevant for the cloud homogeneity.
Line 137: using a SMPS; Give the type of CPC used.
Line 139: Justify why this background may be negligble
Line 140-141: Explain what flushing cycle you mean?
Line 148: 'one'?; What was the temperature homogeneity?
Line 149: What is nearly 85-95%? How reproducible and how controlled can this be adjusted?
Lines 151-160: Give a better explanation of the individual steps and how controlled they can be conducted. Is there any particle generation associated with the steam injection?
Line 171: Why were different flows used to generate the seed particles for dark and illuminated conditions? Why did you dry the seed particles before injection to a chamber with RH>90%?
Line 173: How reproducible was the particle injection and if the target was between 6000-8000 cm-3 why were all experiments reported conducted with lower numbers?
Lines 173-175: Did you use the data from the AAC?
Line 179: Does the Humicap hygrometer really work in vacuum?
Line 180: Give the time resolution of the measurements for all temperature and humidity sensors.
Line 184: What is the reason for the relative low accuracy of the temperature sensors? Could it be improved by calibration?
Line 188: How were the thermocouples shielded from radiative heating?
Line 190: How far from the wall?
Line 191: How was the dewpoint mirror connected to the chamber (heated tube? Tube dimensions) and what was its sample flow and time resolution. What are the distances from the sampling tube inlet and the sensor T6?
Lines 198-199: Add the dimensions and orientation of the sampling tube for the Welas. What were typical sampling losses?

Lines 204-206: Was the SMPS operated with humidified sheath air?

Lines 245-246: Rephrase: E.g. CCN weren't measured directly with a CCN counter.

Lines 247-248: You seem to use only data of this sensor. Give a justification why this sensor was more 'sensitive' than the others in the same chamber?

Line 250: the cloud droplet number

Lines 252-253: Please rephrase. This formulation is irritating. Make always clear what was measured and what was calculated.

Section 3.1: This section should include how the uncertainties of the CFP were determined.

Line 283: Could this be caused by condensation of water on the sensor?

Line 288: Do you mean cloud droplets instead of wet seed particles?

Lines 289-291: Give possible reasons for a bimodal droplet distribution. It would be nice to see the droplet size distribution in a droplet number over size plot.

Line 306: Can you give potential explanations for the changing total temperature drops?

Figure 3: Could be merged with figure 5. Give indicators for the uncertainties of the measured and calculated parameters on the figure.

Table 1: You may add the RH from the Humicap sensor for all experiments aside the one calculated based on T6 + dewpoint mirror.

Line 326: The calculated dry seed particle size.

Line 331: double check the value of 2-3 days for the lifetime (1/e)

Lines 334-336: Give the factors the activated fraction depends on.

Lines 341-346: Isn't this the time needed to increase the RH to the critical supersaturation?

Line 347: avoid 'tiny'

Line 349: Table 1 shows 102.4% RH and here you give 99% for N-IC-1?

Line 352: Indeed, the supersaturation in a dynamic expansion can be higher.

Lines 353-355: Can you give the reader an idea of the turbulences in the chamber? To what extend could the fluctuations shown in the calculated supersaturations based on fluctuating input parameters?

Lines 356-363: The comparisons here are not meaningful without a detailed comparison of the expansion parameters.

Lines 369-370: The chemistry is most likely not affected but the initial cloud activations preferentially activate the most active CCN which then can get lost with a higher probability. Hence, each subsequent cloud activation should require a little higher supersaturation for activation as the most active CCN were removed, or?

Lines 378-379: Does this mean that at the given pumping speed no higher supersaturations could be achieved if starting at lower RH values?

Lines 382-383: replace lower with smaller particle size.

Table 2: Explain the values in brackets and how the uncertainties were calculated.

Figure 4: Consider adding the measured 'wet' seed aerosol size distribution and the calculated wet seed aerosol size distribution before and after cloud.

Lines 441-443: How large is the impact of the chamber inhomogeneities on the cloud properties determined?

Lines 483-491: You may show by comparing all temperature measurements the larger temperature and humidity gradients for the experiments with irradiation.

Line 488: …air is not an efficient light…

Line 508: Quantify the reproducibility of the expansion cooling and cloud activation.

Lines 529-530: Did you measure or calculate a decreasing light intensity in the clouds?